# Discovering plasticity rules that organize and maintain neural circuits

**David Bell**
Department of Physics
University of Washington
Seattle, WA 98195
davidgbe@uw.edu

**Alison Duffy**
Department of Physiology and Biophysics
University of Washington
Seattle, WA 98195

**Adrienne Fairhall**
Department of Physiology and Biophysics
University of Washington
Seattle, WA 98195

## Abstract

Intrinsic dynamics within the brain can accelerate learning by providing a prior scaffolding for dynamics aligned with task objectives. Such intrinsic dynamics should self-organize and self-sustain in the face of fluctuating inputs and biological noise, including synaptic turnover and cell death. An example of such dynamics is the formation of sequences, a ubiquitous motif in neural activity. The sequence-generating circuit in zebra finch HVC provides a reliable timing scaffold for motor output in song and demonstrates a remarkable capacity for unsupervised recovery following perturbation. Inspired by HVC, we seek a local plasticity rule capable of organizing and maintaining sequence-generating dynamics despite continual network perturbations. We adopt a meta-learning approach introduced by Confavreux et al, which parameterizes a learning rule using basis functions constructed from pre- and postsynaptic activity and synapse size, with tunable time constants. Candidate rules are simulated within initially random networks, and their fitness is evaluated according to a loss function that measures the fidelity with which the resulting dynamics encode time. We use this approach to introduce biological noise, forcing meta-learning to find robust solutions. We first show that, in the absence of perturbation, meta-learning identifies a temporally asymmetric generalization of Oja's rule that reliably organizes sparse sequential activity. When synaptic turnover is introduced, the learned rule incorporates an additional form of homeostasis, better maintaining sequential dynamics relative to other previously proposed rules. Additionally, inspired by recent findings demonstrating plasticity in synapses from inhibitory interneurons in HVC, we explore the role of inhibitory plasticity in sequence-generating circuits. We find that learned plasticity adjusts both excitation and inhibition in response to manipulations, outperforming rules applied only to excitatory connections. We demonstrate how plasticity acting on both excitatory and inhibitory synapses can better shape excitatory cell dynamics to scaffold timing representations.

## 1   Introduction and related work

How computational structures are organized and maintained within the brain is a central question within neuroscience. While feedback is clearly essential for learning, self-organization of neural circuits can unfold without feedback, e.g. during development. Brains have evolved specific cell types with nonrandom spatial organization, plasticity rules, and connectivity that likely introduce a

38th Conference on Neural Information Processing Systems (NeurIPS 2024).

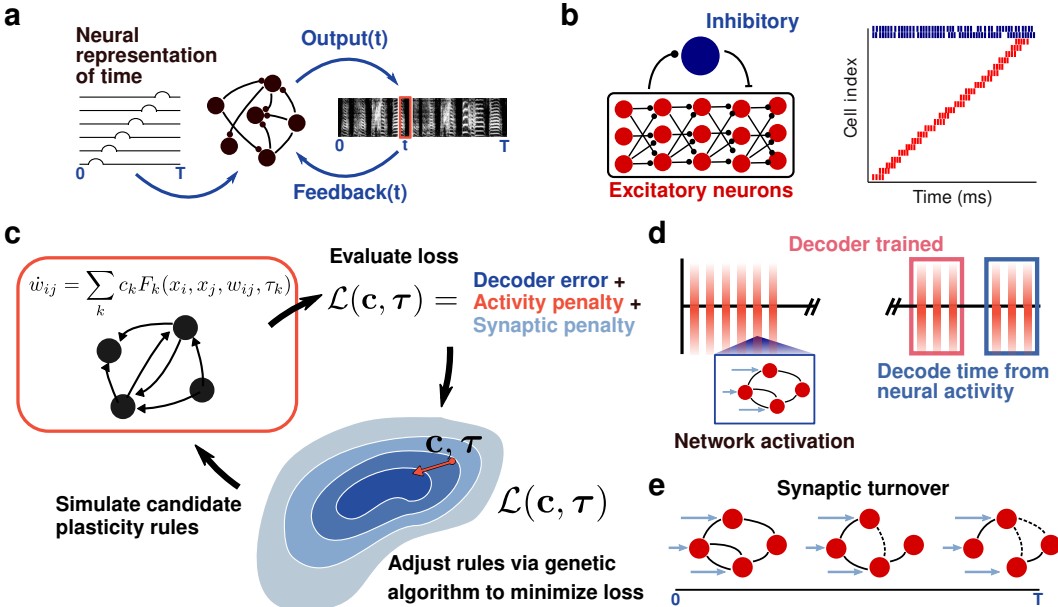

Figure 1: **Meta-learning approach to discovering plasticity rules that organize sequences (a)** In zebra finch song learning, a neural representation of time (left) in HVC simplifies the sequential motor learning task of producing the correct spectral output. **(b)** Putative network structure of zebra finch HVC: a feed-forward, excitatory network with recurrent inhibition (left). HVC excitatory neurons (red) fire sparsely in time while interneurons (blue) fire tonically (right). **(c)** Strategy for learning plasticity underlying sequence organization: candidate plasticity rules, parameterized by a set of coefficients and time constants, are simulated. A loss function is evaluated on the resulting dynamics, and new candidate rules are generated. **(d)** Test procedure for representation of time. Networks are activated 400 times (red bars). From the final 50 activations, six are chosen to train a decoder and six to test the representation by decoding time from neural activity. **(e)** Discovery of robust plasticity rules is encouraged by introducing synaptic turnover, the stochastic addition and removal of synapses, into simulations.

strong set of inductive biases on the information processing they perform. How might organization of useful circuit dynamics be established and maintained throughout life without the need for feedback? Recent work suggests self-organized computations, once established, can accelerate learning and improve performance when experience is limited: Nicola and Clopath [1] demonstrated that a stable high dimensional time signal could improve a network's performance on sequential motor tasks (Fig. 1a). Najarro and Risi [2] learned Hebbian plasticity that orchestrated spontaneous walking behavior in quadruped agents; similar work has shown architectural priors increase the sample efficiency and generalization of RL approaches to locomotion [3, 4]. Additionally, in RL settings, supplying agents with a time input permits them to adopt time-dependent policies [5]. The ability of computational primitives, such as timing representations, to self-organize is challenged by the shifting structure of neural circuits. Synaptic loss, synaptogenesis, cell death, and neurogenesis pose challenges for all learning algorithms, but particularly for self-organization which must be based solely on local information rather than global task performance.

Here, we aim to find plasticity rules that self-organize and maintain one useful computational primitive: sparse, sequential activity. Such activity is widely seen in many areas of the brain including hippocampus [6], cortex [7], and basal ganglia [8]. In the songbird zebra finch, area HVC (used as a proper noun), a cortical-like region, displays sequential activity representing time [9], reducing the problem of motor learning to driving the correct motor neuron at the correct moment [10]. Extensive literature has explored how such sequence-generating circuits could emerge in the absence of feedback [11–17], but has largely focused on either how these structures organize or how they self-maintain, using guessed plasticity rules, and neglecting the effects of ongoing synaptic noise. Further, previous work on sequence organization within HVC has focused on plasticity between excitatory (E) neurons. Recent experimental findings show unsupervised recovery of HVC dynamics is accompanied by changes in both E→E and also inhibitory-to-excitatory (I→E) synaptic strength [18].

Here, instead of imposing a guessed rule, we ask which self-supervised plasticity rules can organize and maintain sequential dynamics within a network. We employ meta-learning, a supervised method to learn learning rules [19–23], in order to discover rules that self-organize a sequence. Our approach stems from a rich history of learning local plasticity rules, including rules that extract representations from data [24], enhance artificial agent performance on familiarity and navigation tasks [2, 25], and explore biologically-plausible replacements or complements to backpropagation [19, 20, 26]. In this study, we parameterize the space of plasticity rules with a basis of activity- and synapse size-dependent terms. The set of coefficients weighting these terms and associated time constants are adjusted to minimize a loss function. Inspired by the HVC context, we pose the loss in terms of the accuracy with which the time since an initial network input can be decoded from the circuit dynamics. We first consider E→E rules alone, and then add I→E and E→I plasticity. We then introduce perturbations to the circuit and investigate which learned plasticity rules promote circuit stability. We find that meta-learned rules for self-organized sequence generation and maintenance contain distinct forms of spike-timing dependent plasticity, homeostasis, and network bounds, which outperform previously proposed sequence-organizing rules in the presence of noise and that plasticity on reciprocal connectivity to inhibitory neurons confers additional stability on the network dynamics. Our main contribution is the exploration of unsupervised and unrewarded plasticity via meta-learning that organizes and maintains a specific and biologically relevant computational motif, a sequence.

## 1.1 Background on zebra finch physiology

In the zebra finch, nucleus HVC contains excitatory neurons that fire sparsely (typically in one burst of spikes) during song and are purportedly arranged in a feed-forward structure (Fig. 1b) [27]. A subset of these cells, known as $HVC_{(RA)}$ neurons, project to downstream nucleus RA (robust nucleus of the archistriatum), which in turn projects to vocal neurons of the syrinx and to the brainstem, which regulates respiration [9]. HVC receives excitatory projections from nucleus Uva, which controls the onset of song syllables [28] and provides input for the duration of song [29]. $HVC_{(RA)}$ neurons inhibit each other disynaptically via a population of inhibitory interneurons [30]. Remarkably, singing behavior can persist when the nucleus is transected [31], demonstrating its resilience.

## 2 Results

### 2.1 Learning biologically plausible plasticity on E→E synapses that organizes a sequence-generating circuit

We first learned a plasticity rule on E→E synapses that organizes a randomly connected network into a sequence-generating circuit in the absence of any perturbation. We initially constrained plasticity to E→E synapses to compare with previously proposed rules, which have largely only considered E→E plasticity. We did not include rules that imposed hard bounds on the size of individual synapses or on the collective strength of all synapses onto a neuron, as we aimed to learn plasticity rules that could permit flexible rescaling of connections in response to perturbations, as has been observed experimentally [18].

We use a network of 25 E and 8 I threshold-linear neurons. Each neuron fired according to $x_j(t) = [V_j(t) - b]^+$, where $V_j(t)$ evolves via $\tau_m \dot{V}_j(t) = -V_j(t) + \sum_i w_{ij} x_i(t)$. Here, $w_{ij}$ is the weight of the synapse $i \rightarrow j$, $\tau_m$ is the membrane time constant, and $b$ is the bias. Initial connectivity (Fig. 2d) was random, but contained no I→I connectivity as is the case in HVC [30, 32] (see Supp. Sec. 1 for all network model details).

#### 2.1.1 Meta-learning procedure

We adopt a meta-learning approach pioneered by Bengio et al. [19] and extended by Confavreux et al [33]. We parameterize a set of plasticity rules with coefficients $c_k$ and time constants $\tau_k$, such that individual synapses, $w_{ij}$, evolve according to

$$\dot{w}_{ij}(t) = \Theta(|w_{ij}(t)|) \sum_k c_k F_k(x_i(t), x_j(t), w_{ij}(t), \tau_k) \tag{1}$$

where $x_i$ and $x_j$ are pre- and postsynaptic activities, respectively, $\Theta$ is the Heaviside function, and $F_k$ is the $k^{th}$ term in the plasticity rule (see Fig. 3a or Supp. Sec. 2 for all terms). The basis includes

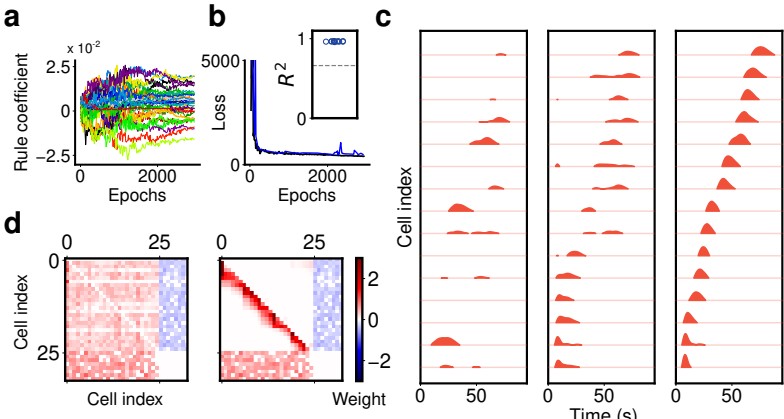

Figure 2: **Meta-learning discovers unsupervised local plasticity rules that organize sequential activity (a)** Evolution of coefficients and time constants during meta-learning. **(b)** Training loss (black) and test loss (blue) during meta-learning. Inset: median decoding accuracy for 14 learned rules across 100 networks (blue points) compared to no plasticity (dashed gray line). **(c)** Meta-learned plasticity rules generate sequence dynamics. Network activity during the first activation (left), $200^{th}$ (middle), and $400^{th}$ (right), sorted by ordering of mean firing time of the final activation ($425^{th}$). Note: plasticity rule is held fixed during simulation. **(d)** Weight matrices at activation 1 and $400$, sorted based on mean firing time of the final activation. Connectivity between E cells organizes into a feedforward structure.

terms that filter pre- and postsynaptic activities with decaying exponentials (denoted e.g. $\tilde{x}_i$) as these terms convey information about the durations of activations and relative ordering, not just their instantaneous rates. Synapses evolving under Eq. 1 were bounded so that they obeyed Dale's law.

We next define a loss function that evaluates the quality of the sequential dynamics organized under a chosen $\{c_k, \tau_k\}$ and attempt to minimize it using an evolutionary strategy, Covariance Matrix Adaptation (CMA-ES) [34]. We use CMA-ES to sample from the space of possible $\{c_k, \tau_k\}$ and evaluate the loss at each point by simulating 10 randomly initialized networks under the given rule and evaluating the resultant dynamics at the end of the simulation. Each simulation is divided into 400 activations of 110 ms. At $t = 10$ ms of each activation, a single fixed neuron is driven by a strong kick of excitation [17]. Following this, all other neurons in the network receive Poisson distributed input for a period of 65 ms. A fraction of this Poisson input is held fixed from trial to trial, mirroring input to HVC from the nucleus Uva, which likely does not provide a fully stochastic signal to the downstream area, [28]. The total loss for a given rule is simply the sum of losses across each of the 10 networks.

### 2.1.2 Loss function

To learn sequences, we define a loss function based on three principles: (1) elapsed time since initial network activation should be readily decoded from network activity, (2) total network activity should be sparse, and (3) total synaptic change in the network should be minimized. The latter two principles impose the assumption that effective plasticity does not require excess network activity or synaptic change to stabilize network function. Our loss is

$$\mathcal{L}(\mathbf{c}, \boldsymbol{\tau}) = \mathcal{L}_{\text{dec}}(\mathbf{c}, \boldsymbol{\tau}) + \lambda_{\text{a}} \sum_i \int_0^T x_i(t) dt + \lambda_{\text{s}} \mathcal{P}(\mathbf{c}, \boldsymbol{\tau}), \tag{2}$$

where

$$\mathcal{P}(\mathbf{c}, \boldsymbol{\tau}) = \sum_{i,j,k} \int_0^T |c_k F_k(x_i(t), x_j(t), w_{ij}(t), \tau_k)| \Theta(|w_{ij}(t)|) dt. \tag{3}$$

$\mathcal{P}(\mathbf{c}, \boldsymbol{\tau})$ penalizes the all synaptic changes due to each component of $\mathbf{F}$. This $L_1$-like penalty on $\mathbf{F}$ penalizes each term not by the size of the term's coefficient, $c_k$, but by the quantity of synaptic change it evokes. While penalizing the magnitude of $c_k$ is standard [20, 33], we take this approach to compare different terms on a common scale, as each component of $\mathbf{F}$ has differing dependence on $x_i$, $x_j$, and $w_{ij}$. $\lambda_{\text{a}}$ and $\lambda_{\text{s}}$ are positive constants weighting the activity and synaptic change penalties.

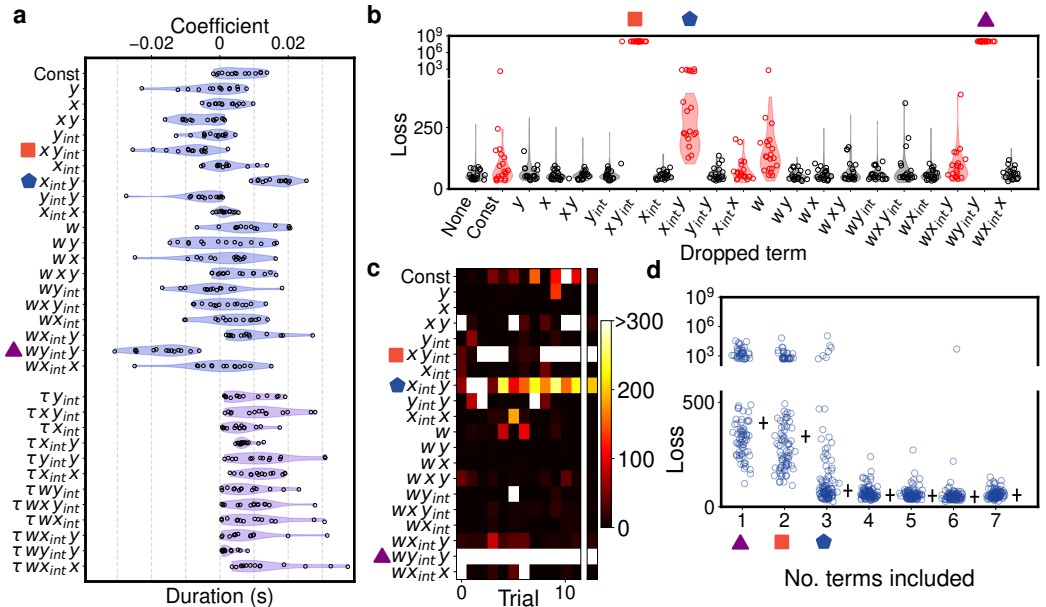

Figure 3: **Perturbing learned plasticity rules reveals dependence on temporally asymmetric Hebbian learning and a postsynaptic activity bound** **(a)** Distributions of coefficients (blue) and time constants (purple) of basis terms across N=14 training instances (individual instances in black). The basis set consists of functions of pre and post-synaptic neural activity and synaptic weights. $x =$ activity of pre-synaptic neuron; $y =$ activity of post-synaptic neuron; $w =$ weight of synapse; $x_{int} = \tilde{x}$ ($x$ filtered with $\tau$). **(b)** Test loss when individual coefficients are set to zero. **(c)** Difference in median loss between full learned solutions run on 100 test networks and solutions with one term dropped. Separate column denotes median loss for terms across all trials. **(d)** Progressive refitting of model in order of impact on median loss with and without term. Symbols indicate the term added at each refitting. Losses for 100 test networks shown (blue); medians shown as black crosses.

To determine the decoder loss $\mathcal{L}_{\text{dec}}(\mathbf{c}, \boldsymbol{\tau})$, the activity of networks was sampled at 500 time points of six activations of the network. From these, a linear decoder was constructed and used to decode the activity at 200 time points of six subsequent activations. Note that since the initial kick of excitation was time-locked to beginning of each activation, constructing a decoder to read out time elapsed since the beginning of the activation was equivalent to decoding time elapsed since the initial excitatory kick was presented.

### 2.1.3 Learned E→E plasticity induces sequences using temporally asymmetric Hebbian learning and a peak postsynaptic activity bound

Meta-learning reliably found plasticity rules that organized random E synaptic connectivity into feed-forward structures that generated sequences when activated (Fig. 2d-e). The organized structures were not grouped into links, as in a synfire chain, but were better described by a kernel in which the strength of a synapse between two cells depended on the lag between their mean firing times (Fig. 2d, right). The set of rules discovered by meta-learning are not sparse in the space of $\{c_k, \tau_k\}$ (Fig. 3a). To test whether learned plasticity rules truly required all terms with nonzero coefficients, we compared 'dropout' variants of the discovered rule, in which one coefficient within $\{c_k\}$ was set to zero, to its unaltered form. We computed the loss of these variants on test networks to determine if a term's absence impacted the loss (Fig. 3b). Computing the change in the median loss of networks organized by the learned solution and a dropout variant across 14 discovered rules revealed reliable trends in the importance of various terms (Fig. 3c). We refit these terms progressively in order of the impact on loss across all training runs and found a sharp elbow at 3 terms: a temporally asymmetric Hebbian learning term, $\tilde{x}_i x_j$ (blue pentagon in Fig. 3), its complement $x_i \tilde{x}_j$ (orange square), and a term second order in the postsynaptic activity multiplied by the synapse size, $w_{ij} \tilde{x}_j x_j$ (purple triangle). The first term was consistently learned with a positive coefficient while the latter two were

always almost negative (Fig. 3a), rendering the effective learning rule

$$\dot{w}_{ij} = c_0 \tilde{x}_i x_j - c_1 x_i \tilde{x}_j - c_2 w_{ij} \tilde{x}_j x_j. \tag{4}$$

where all $\{c_i\}$ are non-negative and time constants are different for each term. The 3 most important terms can be interpreted as a temporally asymmetric generalization of Oja's rule in that the Hebbian learning term, $x_i x_j$, is replaced by the first two terms in Eq. 4, which depend on the relative timing of $x_i$ and $x_j$. The time constant of the third term was on average short ($\sim$1 ms), making this term akin to $w_{ij} x_j^2$, the normalizing term of Oja's rule.

## 2.2  Biological noise alters the learned plasticity rules

We next asked whether ongoing disruptions to network structure alter which plasticity rules are meta-learned. To explore this, we introduced synaptic turnover to the simulation phase of the meta-learning loop. Synaptic turnover is a stochastic process by which existing synapses disappear and new, small synapses emerge (Fig. 1e). Prior to each network activation, all connections were updated according to

$$w_{ij} \leftarrow \begin{cases} 0 & |w_{ij}| > 0 \quad \text{and} \quad x_{\text{ST}} < p_{\text{ST}} \\ w_{ij} & |w_{ij}| > 0 \quad \text{and} \quad x_{\text{ST}} \geq p_{\text{ST}} \\ \epsilon & |w_{ij}| = 0 \quad \text{and} \quad x_{\text{ST}} < p_{\text{ST}} \\ 0 & |w_{ij}| = 0 \quad \text{and} \quad x_{\text{ST}} \geq p_{\text{ST}}, \end{cases} \tag{5}$$

where $x_{ST} \sim U[0,1]$, $p_{ST}$ is the probability of single synapse turnover per activation, and $\epsilon$ is a small positive (negative) constant if the presynaptic cell is excitatory (inhibitory). Since plasticity is unable to act on connections of size 0 (see Eq. 1), synaptic turnover determines the set of synapses available to the plasticity rule. To ensure learned rules were robust to a spectrum of rates of synaptic turnover, only half the networks used to evaluate the batch loss underwent this process.

We found that meta-learned rules were able to organize persistent representations of time despite synaptic turnover, with performance near that of rules learned on unperturbed networks. Our term-sensitivity analysis (Fig 4a) showed that solutions again heavily depended on temporally asymmetric Hebbian learning, i.e. $\tilde{x}_i x_j$ (light blue hexagon), and the bound on postsynaptic activity, $w_{ij} \tilde{x}_j x_j$ (purple triangle); however, we frequently found dependence on two additional terms: one that constantly strengthened all synapses (dark blue pentagon) and an activity bound independent of synapse size (orange square). Refitting the plasticity rule in order of impact on loss demonstrated that these 4 terms recapitulated most of the success of the learned solutions. The effective rule may be written as

$$\dot{w}_{ij} = c_0 + c_1 \tilde{x}_i x_j - c_2 \tilde{x}_j x_j - c_3 w_{ij} \tilde{x}_j x_j, \tag{6}$$

where again all $\{c_i\}$ are non-negative. Synapses for which postsynaptic activity, $x_j$, remains chronically small will be potentiated by $c_0$; however, adding this potentiating term also necessitates an activity bound that does not scale with synapse size, such as the term with prefactor $c_2$ in Eq. 6. To understand this impact of this term, consider a synapse between two neurons whose typical firing times are far apart, i.e. $\tilde{x}_i x_j$ is nearly zero, the fixed point under this rule is

$$\langle w_{ij} \rangle = \max \left( \frac{c_0}{c_3 \langle \tilde{x}_j x_j \rangle} - \frac{c_2}{c_3}, 0 \right) \tag{7}$$

if $\langle \tilde{x}_j x_j \rangle > 0$, where $\langle \cdot \rangle$ denotes the time average. Thus, a large enough choice of $c_2$ prevents every synapse in the network from growing, enforcing sparsity and decreasing the risk of a neuron changing its firing time upon loss of its original inputs. We additionally note that Eq. 6 does not contain $-x_i \tilde{x}_j$, which appeared in the reduced rule learned in the absence of synaptic turnover (Eq. 4). This may be because the roles of $-x_i \tilde{x}_j$ and $-\tilde{x}_j x_j$ are partially redundant: both terms can suppress synapses that run counter to the sequential dynamics in the network. When constant potentiation of all synapses occurs, the term in $\tilde{x}_j x_j$ is preferable as it offsets constant potentiation of all synapses. In the unperturbed context, where constant synaptic growth is unnecessary, the term in $\tilde{x}_j x_j$ is problematic in that it can set all afferent synapses to a driven neuron to zero (whereas $-w_{ij} \tilde{x}_j x_j$ cannot). Thus, the term in $x_i \tilde{x}_j$ becomes preferable in the unperturbed context.

### 2.2.1  Comparison to existing models of sequence formation

Do these discovered learning rules more robustly encode timing than previously proposed rules when the circuit is disrupted with biologically relevant noise? We hypothesized this would be true given

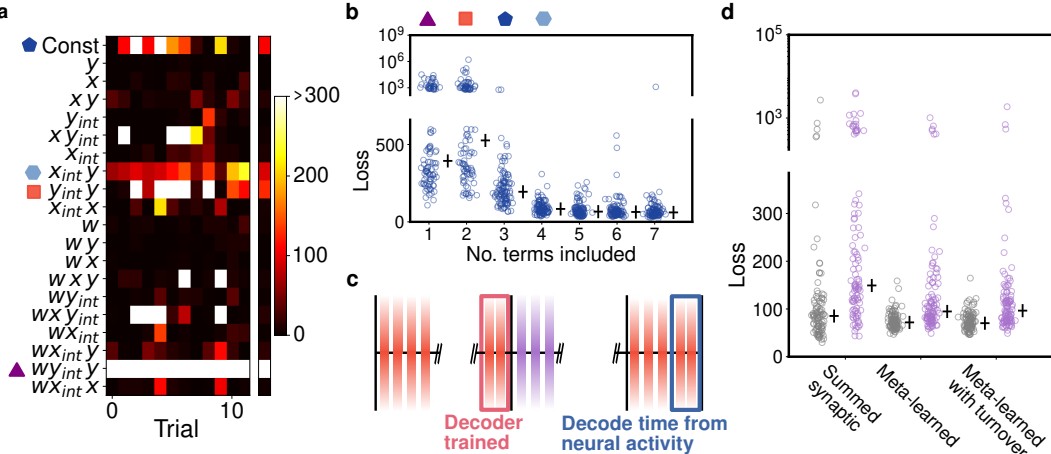

Figure 4: **Discovered rules organized dense feedforward structures (a)** Median impact of loss when coefficient of given plasticity is set to zero for a given learned solution. **(b)** Refitting terms in order of median impact on loss shows sequence generation in synaptic turnover context is well captured by 4 terms. **(c)** Task structure for comparison between different rules. Rules are given 400 activations to organize dynamics. At the end, the decoder is constructed. Networks then undergo synaptic perturbation for 150 activations. Finally, the decoder attempts to decode time from the resulting dynamics to determine the loss. **(d)** Comparison between multiplicative Hebbian learning and summed synaptic bound, meta-learned without synaptic turnover, and meta-learned with synaptic turnover on time encoding task illustrated in (c). Trials shown in grey do not include synaptic turnover; light purple include synaptic turnover. Black crosses indicate median values for each condition.

that discovered rules do not impose hard bounds on the size of single synapses, total synaptic strength onto a neuron, nor number of synapses, as other models of sequence formation have [11–13, 17]. The absence of these constraints permits compensatory rescaling of synapses in response to disruptions. We compared meta-learned rules trained with and without synaptic turnover to a previously proposed sequence learning rule that used multiplicative asymmetric Hebbian learning, a single synapse bound, and bounds on the total strength of synapses onto and out of individual neurons [11]. Each rule was applied to 100 test networks for 400 activations without disruption. Following this, a decoder was constructed to read out time from the neural activity, and then synapses in the networks were turned over during an additional 150 activations, after which the loss was evaluated using the constructed decoder (Fig. 4c). We compared the best versions of each rule when synapses were turned over (with probability $p_{ST} = 0.00072$) at each activation, equivalent to a 90% probability of individual synapse survival during the disruption period (Fig. 4d, light purple points), and when they were not turned over (Fig. 4d, grey points; see Supp. Sec. 4). We used the Kruskal-Wallis $H$ test to test for equality of medians. We found that meta-learned rules trained with and without synaptic turnover outperformed the rule based based on rigid synapse constraints ($p = 2.5 \times 10^{-8}$, Cohen's $d = -0.45$ and $p = 7.2 \times 10^{-7}$, Cohen's $d = -0.44$, respectively), while in the absence of perturbation, medians were not distinct after 4-fold Bonferroni correction ($p = 0.016$, Cohen's $d = -0.29$ and $p = 0.029$, Cohen's $d = -0.30$, respectively). When we studied the connectivity structure of networks organized by meta-learned rules, we found that the discovered rule generated denser feedforward connectivity in comparison to other plasticity rules with alternative forms of Hebbian learning and heterosynaptic competition (see Supp. Sec. 5).

## 2.3 Including inhibitory plasticity

Meta-learning allows the exploration of multiple plasticity rules operating on distinct sets of synapses within the same circuit, as might arise if there are multiple cell types [35, 36]. In particular, I→E plasticity has been the focus of much recent work [37–42]. The interaction of many plasticity rules is challenging to analyze theoretically, but meta-learning allows exploration of these interactions [43, 44]. Recent evidence suggests there may be multiple forms of plasticity within sequence-generating circuits: Wang et al. [18] increased the intrinsic excitability of *in vivo* HVC$_{(RA)}$ neurons and found that the strength of both E→I and I→E connections could dynamically shift in response. Targeted cells received increased total inhibitory synaptic strength and decreased excitatory strength.

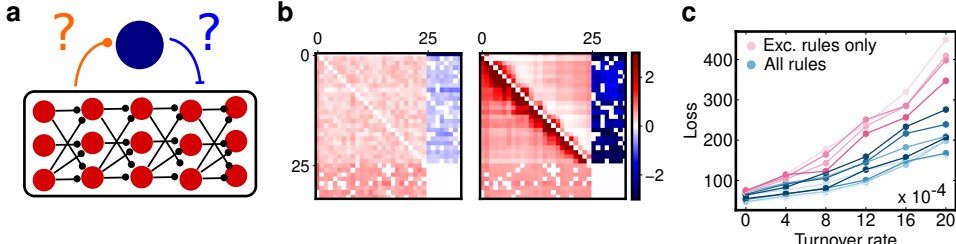

Figure 5: **Learning plasticity on all synapses** **(a)** Schematic of synapses upon which we newly allow plasticity. **(b)** Weight matrices on activations 1 and 400 of a network evolving under plasticity rules learned on all 3 sets of synapses. I synapses increase. **(c)** Comparison of performance of rules learned only on E→E synapses (red) (N=5) versus all sets of synapses (blue) (N=8) when training includes synaptic turnover across varying rates of synaptic turnover.

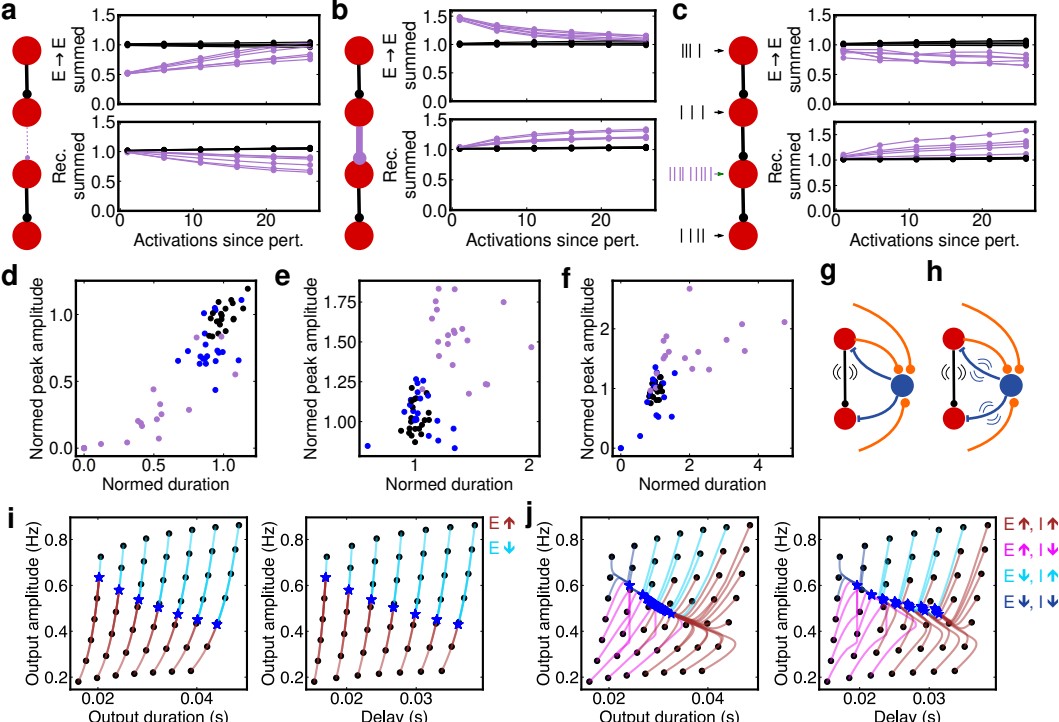

Figure 6: **Network perturbations reveal homeostatic compensation in E→E and I→E synapses** **(a)** Response of the magnitude of summed E→E weights $|W_i^{\text{exc}}| = |\sum_k w_{k,i}^{E\to E}|$ (top) and summed recurrent weight $|W_i^{\text{rec}}| = |\sum_k w_{i,k}^{E\to I} w_{k,i}^{I\to E}|$ (bottom) to the imposed scaling down of E→E weights to a single cell. Values for neuron with perturbed inputs shown in light purple; all others black. Each line represents average values for a single learned rule over N=20 networks. **(b)** Same as (a), but for imposed upscaling of E→E weights to one cell (light purple). **(c)** Same as (a), but frequency of stochastic inputs to one cell is greatly increased (light purple) relative to all other cells (black). **(d-f)** Postsynaptic response of affected neurons in 20 networks plotted in the space of normalized peak amplitude and duration for manipulations (a-c). Black dots represent pre-perturbation responses; light purple, immediately after perturbation; blue, 25 activations after perturbation. **(g)** Schematic of feed-forward motif maintaining homeostasis via its E input. **(h)** Schematic of feed-forward motif with homeostasis on E→E and I→E synapses. **(i)** Phase flow in space of output firing envelope duration and amplitude (left) for a single neuron. Black points are first responses of the neuron for various inputs; blue stars represent responses after 200 activations. Note: initial and final output durations and delays are roughly the same. Colors of trajectories indicate how the neuron's input weight changes to satisfy the plasticity rule: see legend. **(j)** If the forms of homeostasis on E→E and I→E maintain different aspects of the postsynaptic response, an attractor forms in the duration and peak amplitude of the postsynaptic response (left). The response delay is also constrained (right).

To understand the role of this additional plasticity, we meta-learned plasticity on all sets of synapses within the circuit (Fig. 5a) both with and without synaptic turnover. Specifically, we attempted to learn three independent plasticity rules that operated on three distinct groups of synapses (E→E, E→I, and I→E; see Supplementary Section 6).

### 2.3.1 E→I and I→E synaptic plasticity improves decoding of time, particularly in the presence of synaptic turnover

Meta-learning uncovered triplets of plasticity rules that successfully organized initially random networks into sequence generators (Fig. 5b). When trained with turnover in E→E synapses, solutions that acted upon E→I and I→E synapses in addition to E→E outperformed solutions that only acted upon E→E connections, particularly when the rate of synaptic turnover was high (Fig. 5c), suggesting this additional plasticity played an important role in maintaining network dynamics through perturbation.

To investigate how rules acting on all synapses generated improved time representations, we repeated the dropout analysis. We found that E→E plasticity within these learned triples was largely similar to the rules previously learned on E→E synapses alone (Eqs. 4 and 6): solutions were consistently sensitive to the removal of $\tilde{x}_i x_j$ and $w_{ij}\tilde{x}_j x_j$, which appeared consistently with positive and negative coefficients, respectively. Further, dependence on these terms persisted when we trained networks with turnover on E→E synapses or I→E synapses (Supp. Fig. 6). As expected, we also found that solutions depended heavily upon terms that acted upon E→I and I→E synapses, but this plasticity was more difficult to interpret due to increased trial to trial variability in the discovered rules. We found, however, that the E→I plasticity rule consistently depended on the second order presynaptic term $\tilde{x}_i x_i$, which always appeared with a positive coefficient, suggesting that E cells project to inhibitory counterparts with a strength that increases with the E cell level of activity. An implication of dependence on this term is that the strength of an E neuron's recurrent inhibition, defined as $|W_i^{\mathrm{rec}}| = |\sum_k w_{i,k}^{\mathrm{E}\to\mathrm{I}} w_{k,i}^{\mathrm{I}\to\mathrm{E}}|$, where $i$ is the index of the E cell and $k$ indexes the I cells to which it projects, might depend on its level of activity. Thus, ablation of excitatory inputs to an E cell might cause its recurrent, disynaptic inhibition to lower in a manner that homeostatically restores its firing.

### 2.3.2 Recurrent inhibition of E neurons is homeostatic in networks with learned plasticity

As the role of E→I and I→E plasticity was not completely clarified by our perturbations so far, we next investigated how this plasticity adjusted synapses coupled to an individual E cell when its typical input was manipulated. Noting the dependence of the E→I plasticity rule on $\tilde{x}_i x_i$, we hypothesized that a targeted neuron's recurrent inhibition and excitatory afferents might be adjusted in concert to restore its typical firing pattern. We used discovered learning rules to organize sequences and then performed three varieties of *in silico* manipulations of individual E cells within these networks. In the first, we scaled down the excitatory afferents to the targeted E cell by 50% (Fig. 6a, diagram). In the second, we scaled up the same connections by 50% (Fig. 6b). In the last, we increased the rate of the targeted cell's input Poisson process by a factor of 10 (Fig. 6c). This final manipulation mirrored the viral insertion of NaChBac to $\mathrm{HVC_{RA}}$ neurons in Wang et al., which causes these cells to become hyper-excitable [18, 45, 46]. Wang et al. found that manipulated cells recruited additional inhibition and weakened excitatory afferents.

We found that scaling down a targeted E cell's excitatory afferents resulted in a rescaling of those excitatory weights and an accompanying decrease in $|W_i^{\mathrm{rec}}|$ (Fig. 6a). We further found that these synaptic changes restore the initial firing pattern of the E cell: in Fig 6d, we plot the initial responses of targeted E cells across 20 self-organized sequences in the space of duration and peak amplitude of response (black points). Scaling down the excitatory afferents initially causes both the peak amplitude and duration of response to decrease (Fig. 6d; light purple points), but these responses largely recover after ∼25 activations of the network (Fig. 6d; blue points). When we instead strengthened E afferents to the targeted cell, we found these connections weakened and recurrent inhibition strengthened in compensatory fashion (Fig. 6b). In these networks, we found that targeted E cell responses that were initially lengthened with increased peak amplitude (Fig. 6e; light purple points) were reduced back to their pre-perturbation values of duration and amplitude (Fig. 6e; black points show initial responses; blue points, responses after 25 activations). Finally, rendering the targeted E cell hyper-excitable caused it to scale down its excitatory afferents and increase its recurrent inhibition in a manner that restored its typical firing pattern. In summary, plasticity rules on excitatory afferents and recurrent

inhibition operate in tandem to maintain the firing pattern of the neuron (see Supp. Sec. 7 for additional details).

### 2.3.3 Two forms of homeostasis create an attractor in postsynaptic response and timing

How might modulation of recurrent inhibition contribute differently to the modulation of excitation in preserving network dynamics? We hypothesized that distinct plasticity on different sets of synapses might confer a robust representation of time if these rules governed distinct aspects of the desired network function. For instance, within our HVC-like circuit, peak firing of E neurons might be controlled by E→E plasticity while total activity might be controlled by I→E plasticity. Since excitatory and inhibitory inputs to E cells differ in their timescales (E inputs are transient while I inputs are relatively tonic), we reasoned plasticity on both sets of synapses should increase the control of the postsynaptic response. While it is already known that multiple plasticity mechanisms can sharpen the responses of neurons to stimuli [37], prior work has not addressed whether similar plasticity might be leveraged to preserve the timing of such responses, which is crucial to timing representations. To explore this, we constructed a simplified model of a single neuron responding to a broad range of excitatory inputs, characterized by varying peak amplitudes and durations, and a tonic inhibitory input. We compared the responses of the neuron when both the excitatory and inhibitory input synapses (Fig. 6h) evolved under plasticity rules to responses produced when only the excitatory synapse evolved under a plasticity rule (Fig. 6g) and found the two rule model was able to better constrain the duration, peak amplitude, and timing of the postsynaptic response the E neuron (Fig. 6i-j; see Supp. Sec. 8 for full description of reduced model).

## 3 Discussion and limitations

Meta-learning plasticity rules via stochastic optimization is a promising technique, but suffers a number of limitations. One, the optimization process becomes expensive as the size of the rule basis, the number of neurons in the network, and amount of simulation time required grows. Further, CMA-ES may require many epochs to converge on good solutions. Training E→E plasticity (plasticity on all synapses) across 10 networks in batch typically required 24 (72) hours of compute on 30 Cascade Lake or Ice Lake Intel CPU cores to yield reasonable solutions. Two, meta-learning tends to generate different solutions based on the seed; due to the expensive nature of each trial, we did not carry out enough trials to claim full knowledge of the solution space. Three, the plasticity rules learned were quite dense in our choice of basis, limiting interpretation, and we ultimately employed perturbations to better understand the critical terms. Four, the choice of basis limits the space of discoverable rules; for instance, we did not include feedback-modulated plasticity in this study. Five, the initial connectivity of the circuit likely has a strong bearing on the sort of plasticity that successfully can leverage it [47, 48].

Prior to any experience, intrinsic, self-organized dynamics within the brain can serve as powerful priors that can accelerate and shape successful, feedback driven learning. In this work, we study how one such computational primitive could emerge by adapting a meta-learning procedure to learn the learning rules that self-organize and maintain robust representations of time in neural dynamics. Meta-learning discovers a temporally asymmetric (STDP-like) generalization of Oja's rule that organizes and maintains sparse, sequential activity out of initially random connectivity, which outperforms other models of sequence generation in the presence of synaptic turnover by permitting flexible rescaling of inputs to restore dynamics. Additionally we found that plasticity rules learned on all sets of synapses outperform plasticity rules applied only to E connections. Through a toy model, we show how plasticity on all synapses could confer extra timing stability if the plasticity in distinct sets of synapses act on different moments of an E neuron's activity.

In this work, we have developed a paradigm to understand how computational primitives might self-organize within neural circuits and selected sequences as our test example. Future work could study the emergence of other canonical forms of neural dynamics that have been widely identified in brain activity and serve as fundamental components of computation, such as line attractors or limit cycles, or how plasticity rules generating such components interact with rules requiring feedback.

## 4   Acknowledgements and disclosure of funding

We would like to thank Carlos Lois, Zsofia Török, Bo Wang, Patrick Zhang, and Leenoy Meshulam for useful discussions. This work was supported by the Simons Collaboration for the Global Brain and an NIH BRAIN grant (5R01NS104925). This work was facilitated through the use of advanced computational, storage, and networking infrastructure provided by the Hyak supercomputer system and funded by the STF at the University of Washington.

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
