# OpenReview forum: "Discovering plasticity rules that organize and maintain neural circuits"
_NeurIPS.cc/2024/Conference — NeurIPS 2024 poster_

### Official Review · Reviewer_96SN · 2024-07-14

**Soundness:** 3
**Presentation:** 2
**Contribution:** 2
**Rating:** 5
**Confidence:** 2

**Summary:**

This paper introduces meta-learned plasticity rules for sequence generating neural circuits. While existing works demonstrated that the dynamics in HVC is generated by both excitatory and inhibitory synaptic updates, their approaches are based on a guessed rule. Motivated from this, the authors experimentally demonstrate that synapses with meta-learned plasticity rules introduce enhanced stability on the network dynamics under noisy environments, which is also biologically plausible (e.g., homeostasis)

**Strengths:**

- The paper is overall well written and key question the authors are trying to validate (line 33-34) is presented clearly in the introduction.

- The motivation of this paper to understand the self-organizing networks with meta-learned plastic rules is reasonable and biologically plausible.

- In the field of modeling sequence generating circuits in HVC, the author's investigation is novel considering others are mostly based on a relatively simple and guessed rule.

**Weaknesses:**

- Though I appreciate the novel problem statement using meta-learned plasticity rules, I could not find technical novelties in the paper. In other words, my primary concern is if this paper presents any novel insights in machine learning perspective.

- For example, meta-learning of plasticity rules has been widely explored in various tasks [1,2,3]. Some of these papers have already employed the similar optimization techniques such as CMA-ES to quickly optimize the meta-learning parameters. Hence, I think the authors should have discussed these methods in the paper.

[1] Miconi, Thomas, Kenneth Stanley, and Jeff Clune. "Differentiable plasticity: training plastic neural networks with backpropagation." International Conference on Machine Learning. PMLR, 2018.

[2] Najarro, Elias, and Sebastian Risi. "Meta-learning through hebbian plasticity in random networks." Advances in Neural Information Processing Systems 33 (2020): 20719-20731.

[3] Rodriguez, Hector Garcia, Qinghai Guo, and Timoleon Moraitis. "Short-term plasticity neurons learning to learn and forget." International Conference on Machine Learning. PMLR, 2022.

- Also, the idea of homeostasis using inhibitory and excitatory synapses are not new in machine learning perspectives [1,2]. I believe the main contribution in this paper is to understand the meta-learned plasticity rules in modeling neuronal sequences. If so, it seems like applying existing methods to understand another problem. Please correct me if I’m wrong.

[1] Yoshida, Naoto, et al. "Embodiment perspective of reward definition for behavioural homeostasis." Deep RL Workshop NeurIPS 2021. 2021.

[2] Kang, Beomseok, Biswadeep Chakraborty, and Saibal Mukhopadhyay. "Unsupervised 3D Object Learning through Neuron Activity aware Plasticity." arXiv preprint arXiv:2302.11622 (2023).

- It would be nice to have a background section for readers who are not familiar with neuronal sequence modeling and zebra finch HVC.

- The current reference (citation) style looks different to NeurIPS format. I think the references should be written in [x] or (x).

**Questions:**

N/A

---

> ### Author Rebuttal · Authors · 2024-08-07
>
> We thank the reviewer for their comments and for the additional references.
>
> We agree with the reviewer that the novelty of this work primarily lies in the application of an existing technique to a specific question within neuroscience: how might the organization of useful circuit dynamics that accelerate learning be established and maintained without feedback? We argue, however, that the application of this technique to this question is nontrivial. As we argue in the common response, prior studies that employ meta-learning largely operate on feed-forward networks, whose outputs can be rapidly computed, have known solutions within the basis, or operate on a limited basis set (see common response for details and citations). In this work, we show solutions to meta-learning problems can be dense within a provided basis and that coefficient magnitude does not necessarily communicate the contribution of a term to shaping of synapses. We argue this work presents an important case study for other authors seeking to apply meta-learning to questions within neuroscience.
>
> We thank the reviewer for pointing out important work from Miconi et al., Najarro et al., and Rodriguez, et al. We have added these references.
>
> We have additionally added a background section on zebra finch physiology, including a review of areas HVC and RA. This section now reads:
>
> “In the zebra finch, premotor nucleus HVC contains excitatory neurons that fire sparsely (typically in one burst of spikes) during song and are purportedly arranged in a feed-forward structure (Fig. 1c). A subset of these cells, known as $\mathrm{HVC}\_{\rm (RA)}$ neurons, project to downstream nucleus RA (robust nucleus of the archistriatum), which in turn projects to vocal neurons of the syrinx and to the brainstem, which regulates respiration \cite{Hahnloser_2002}. HVC additionally receives excitatory projections from nucleus Uva, which controls the onset of song syllables \cite{Moll2023} and provides input for the duration of song \cite{Danish2017}. $\mathrm{HVC}\_{\rm (RA)}$ neurons inhibit each other disynaptically via a population of inhibitory interneurons within HVC \cite{Kosche_2015}.”
>
> We agree with the reviewer that the idea of homeostasis via modification of excitatory and inhibitory synapses is not new. However, our work suggests exactly how homeostasis might be maintained via these two distinct forms of input in the sequence generation context. We argue this is valuable because our proposal for homeostatic control can maintain not just the shape of an E cell’s firing rate response but also its timing relative to its input. We clarify this point in Sec. 2.3.3. We additionally thank the reviewer for suggesting additional references we missed.
>
> We have also fixed the formatting of the references as the reviewer has suggested.

---

> > ### Comment · Reviewer_96SN · 2024-08-14
> >
> > Thank you for addressing my concerns and clarifying the contribution of this work. I understand that the primary contribution at the perspective of neuroscience. To be honest, I'm not much familiar with neuroscience and still not sure on how significant contribution this paper has made. Considering the answer and evaluation from other reviewers, I would like to increase my final score to 5 but decrease the confidence to 2.

---

### Official Review · Reviewer_6oGR · 2024-07-15

**Soundness:** 3
**Presentation:** 3
**Contribution:** 3
**Rating:** 7
**Confidence:** 3

**Summary:**

The authors investigate the development and persistence of intrinsic connectivity structures within a
neural region without external input. They propose that a local plasticity rule can lead to self-organising connectivity motifs, creating inductive biases beneficial for subsequent information processing. The study focuses on sparse and sequential dynamics, as observed in the HVC region of zebra finches. The paper explores which plasticity rules can organise and maintain these sequential dynamics using meta-learning.The space of plasticity rules is parameterised using terms based on activations and synapse size.
The coefficients and their time constants are adjusted to minimise the loss, which comprises three
components: encouraging sparsity in activation, synaptic change, and improving the accuracy of time decoding. Initially, excitatory-excitatory rules are considered, followed by inhibitory-excitatory and excitatory-inhibitory rules. The study evaluates the efficiency of these rules under synaptic turnover and compares them to established Hebbian plasticity rules. Evolutionary algorithms identified a rule resembling a temporally generalised Oja’s rule, which
outperformed a multiplicative asymmetric Hebbian rule in the context of synaptic turnover.

**Strengths:**

This paper is conceptually significant, presenting the novel idea that local synaptic plasticity
could be crucial for maintaining inductive biases in the form of intrinsically-maintained
computational motifs. These internally generated dynamics must interact with synaptic
changes induced by supervised learning—they need to support this form of learning without
being overwhelmed by it. This perspective introduces a fresh angle to the
understanding of the function and origin of local plasticity rules.
The authors primarily focus on identifying synaptic rules that can create stable inductive
motifs resilient to perturbations. The analyses presented are thorough and convincingly
support this point. The use of meta-learning to estimate these plasticity rules is an innovative
approach. Additionally, the detailed examination of the estimated rule offers an interpretable
outcome from a complex machine learning technique - a rare occurrence in the field.
A particular strength of the paper is its exploration of not only E-E but also E-I and I-E rules.
Furthermore, the authors again analyse the estimated rule in a manner that provides interpretable
outcomes.

**Weaknesses:**

The manuscript occasionally feels unfinished and rushed, particularly in section 2.3.1.
The references to Figure 3 (blue circles, green stars?) and the lack of references to Figure 4
in the text were often frustrating. Additionally, I was keen to understand why the second term
estimated without noise lost its significance when noise was introduced, but the authors did
not address this point.
Furthermore, it was somewhat anticlimactic that the authors did not explicitly explore the
interaction of the estimated plasticity rule with subsequent supervised learning, as this was
strongly hinted at in the introduction. For instance, are networks with the estimated rule better
at reproducing songs than models with a traditional Hebbian rule?
Lastly, I found it odd that the decoding of time was not reported throughout the paper. Since
the reduction of loss can be attributed to three factors, this omission is noteworthy.

**Questions:**

- Why does the second term, which was estimated without noise, lose its importance when noise
is introduced? The authors did not address this point.
- Why did the authors not explicitly explore the interaction between the estimated plasticity rule and
a subsequent supervised learning procedure, despite strongly hinting at this in the introduction?
- Are networks using the estimated rule more effective at reproducing songs than those employing a
traditional Hebbian rule?
- Why was the decoding of time not reported throughout the paper, given that the reduction of loss
could be influenced by three different factors?
- Could you clarify why the estimated rule is considered a temporally generalised Oja’s rule?

**Limitations:**

The authors were aware of the limitations of the methods used in the paper and addressed
them accordingly.

---

> ### Author Rebuttal · Authors · 2024-08-07
>
> We thank the reviewer for their careful evaluation of our manuscript. We have worked to address all comments and rewrite our paper accordingly.
>
> We now provide a rationale for why the second term in Eq. 4 loses its significance in the perturbed context. We have added the following to Sec. 2.2:
>
> “We additionally note that Eq. 6 does not contain $-x_i \tilde{x}_j$, which appeared in the reduced rule learned in the absence of synaptic turnover (Eq. 4). We expect this is because the roles of  $-x_i \tilde{x}\_j$ and $- \\tilde{x}\_j x\_j$ are partially redundant: both terms can suppress synapses that run counter to the sequential dynamics in the network. When synapses are turned over, the latter term is preferable as it offsets constant potentiation of all synapses. In the unperturbed context, constant synaptic growth is unnecessary, and therefore $- \\tilde{x}\_j x\_j$ is problematic in that it can set all synapses to a driven neuron to zero (note $-w\_{ij} \\tilde{x}\_j x\_j$ cannot do this). Thus, $-x\_i \\tilde{x}\_j$ becomes preferable.”
>
> As we note in the common response to all reviewers, we chose to focus on the self-organization of the timing scaffold as the utility of a timing representation to support motor learning has already been well studied (e.g. Duffy et al. PNAS 2019). See common response for full details.
>
> Regarding comparison between the estimated rule and a traditional Hebbian rule, Hebbian plasticity that only potentiates coincident activity (as found in a Hopfield network) cannot be responsible for the organization of sequential activity since asymmetry in the connectivity is required; traditional Hebbian plasticity tends to foster symmetric connectivity (see Sompolinsky and Kanter, 1986). We therefore chose to compare the estimated rule to an existing rule previously used in the literature in the context of this system (Fiete et al., 2010) which employs a temporally asymmetric form of Hebbian learning that is similar to that of our estimated rule, but imposes a sum on the total synaptic strength onto and out of individual neurons. We compare the estimated rule to this pre-existing rule in Fig. 4d.
>
> We have now reported the accuracy of time decoding in Fig. 2. For all rules, the dominant contribution to the loss is the accuracy of time decoding, reported as $1000(1 - R^2).$
>
> We have clarified the sense in which the rule estimated in the unperturbed context on E$\rightarrow$E synapses is a generalization of Oja’s rule. We have added the following to Sec. 2.1.3:
>
> “The 3 most important terms can be interpreted as a temporally asymmetric generalization of Oja's rule in that the Hebbian learning term, $x\_i x\_j$, is replaced by the first two terms in Eq. 4, which depend on the relative timing of $x\_i$ and $x\_j$.”
>
> We note specifically, that if $c\_2$ in Eq. 4 is set to zero and the constants on the first and second terms are made very small and their coefficients are identical, this rule is exactly Oja’s.
>
> We apologize for the lack of clarity in section 2.3.1. We have rewritten it to emphasize the following key points:
> Plasticity rules learned on all synapses outperform rules that only operate on E$\rightarrow$E synapses, particularly when the rate of synaptic turnover is high.
> Perturbing the learned plasticity rules revealed that the E$\rightarrow$E was largely the same as in the prior sections.
> We found E$\rightarrow$I plasticity heavily depended on a term that was second order in the presynaptic activity, $\tilde{x}_i x_i$. It always had a positive coefficient.
> Dependence on this term implied that the synaptic strength of a neuron’s self inhibition might depend on its activity level, suggesting a type of homeostasis.
>
> The last paragraph of 2.3.1 now reads:
>
> “To investigate how rules acting on all synapses generated improved time representations, we repeated the dropout analysis introduced in Section 2.1.3, We found that E$\rightarrow$E plasticity within these learned triples was largely similar to the rules previously learned on E$\rightarrow$E synapses alone (Eqs. 4 and 6): solutions were consistently sensitive to the removal of $\tilde{x}\_i x\_j$ and $w\_{ij} \tilde{x}\_j x_j$, which appeared consistently with positive and negative coefficients, respectively. Further, dependence on these terms persisted when we trained networks with turnover on E$\rightarrow$E synapses or I$\rightarrow$E synapses (Supplement, Fig. 2). As expected, we also found that solutions depended heavily upon terms that acted upon E$\rightarrow$I and I$\rightarrow$E synapses. In general, we found that this plasticity was more difficult to interpret due to increased trial to trial variability than the E$\rightarrow$E plasticity. We did find, however, that the E$\rightarrow$I plasticity rule consistently depended on the second order presynaptic term $\tilde{x}\_i x\_i$, which always appeared with a positive coefficient, suggesting that E cells project to inhibitory counterparts with a strength that increases (perhaps nonlinearly) with their level of activity. An implication of dependence on this term is that the strength of an E neuron's recurrent inhibition, defined as $|W\_i^{\\rm rec}| = |\sum\_k w\_{i, k}^{\rm E\rightarrow I} w\_{k, i}^{\rm I\rightarrow E}|$, where $i$ is the index of the E cell and $k$ indexes the I cells to which it projects, might depend on its level of activity; thus, ablation of excitatory inputs to an E cell might cause its recurrent inhibition to lower in a manner that homeostatically restores its firing.”
>
> We have additionally fixed references to Figs. 3 and 4.

---

> > ### Comment · Reviewer_6oGR · 2024-08-12
> > **Reply**
> >
> > Thank you for addressing all points raised. We will keep our score.

---

### Official Review · Reviewer_4Ykq · 2024-07-19

**Soundness:** 3
**Presentation:** 3
**Contribution:** 3
**Rating:** 7
**Confidence:** 3

**Summary:**

This work applies a meta-learning procedure for plasticity rule discovery to a neural network model of sequence generation. Plasticity rules are constructed from parameterized basis functions, and the parameters are found through evolutionary search based on a fitness function quantifying how accurately the plasticity rules train the network to represent elapsed time within episodic rollouts. The work analyzes the plasticity rules discovered in a variety of conditions: with noise, with synaptic turnover, and with plasticity enabled on excitatory and/or inhibitory neurons.

**Strengths:**

**Significance**: Hand-tuning plasticity rules can be tedious and suboptimal, so automated methods are important for the more efficient and effective design of models. By applying this method to a neural network model of sequence generation (specifically, a model of HVC), this work tests the procedure in a new setting and also develops a more robust plasticity rule for sequence-generating dynamics.

**Novelty**: It seems the novelty is in the application of this method to a HVC model and the evaluation of the learned plasticity rules under a variety of conditions. If there are other contributions, it would be helpful to be clearer about contextualizing this in the related work.

**Technical quality**: This work provided an extensive evaluation of the model in a variety of conditions. The analysis of the discovered plasticity rules was particularly interesting.

**Presentation clarity**: The methods and results are clearly explained.

**Weaknesses:**

**Wiring vs plasticity**: It seems like an assumption of this work is that the sequence generating structure of HVC neural circuits emerges largely due to activity-dependent plasticity in a network with initially random connectivity. What is the role of activity-independent wiring that takes place during development? And how much of an effect does more structured initialization have on the discovered plasticity rules?

**HVC predictions**: As a main contribution of this work is improving a model of HVC, what experimental predictions will result from this computational analysis?

**Figures improvements**: The figures in general have pretty small text, inconsistency in formatting, and subpanels that are cramped. Fig 1 especially has a lot going on, uses a variety of fonts, and adopts a colorblind-unfriendly palette. Fig 2c, I didn't see why the 3 trials were needed; could it just show one and put other trials into the Supplementary? Fig 4d, use 45deg or 90deg xlabel angles instead?

**Text improvements** (minor): (1) The neuron model notation on Line 77 took me a while to parse and seems non-standard according to colleagues; ultimately, it is a simple exponential decay and such neural dynamics are usually expressed in dv/dt notation. (2) The spacing on Page 11 is messed up due to the type of latex linebreak used.

**Questions:**

See above.

**Limitations:**

Yes, the authors adequately address the limitations in the Discussion section.

---

> ### Author Rebuttal · Authors · 2024-08-07
>
> We thank the reviewer for their helpful comments. We have worked to address their suggestions and make changes to the manuscript accordingly.
>
> We agree with the reviewer that activity-independent wiring plays an important role in determining the capacity of a network to self-organize (see Lakshminarasimhan et al, 2024, Litwin-Kumar, 2017). Tyulmankov et al, 2022, also learned Hebbian-like rules via a supervised learning strategy, but first optimized connectivity in their networks via backpropagation. Our study makes a few assumptions about initial connectivity, e.g. we do not interconnect interneurons as this connectivity is very rare in HVC, but we did not explore, for instance, how the sparsity of different types of connectivity might affect the learning rules. We now note this in the discussion section, saying:
>
> “... the activity-independent initial connectivity of the circuit likely has a strong bearing on the sort of plasticity that successfully can leverage it \cite{litwin_kumar, lakshminarasimhan_2024}. For instance, sparsity is known to affect the efficacy of supervised versus unsupervised plasticity. We did not explore this important facet while learning plasticity rules, though in principle it could be included as part of the optimization.”
>
> We have attempted to clarify our contributions beyond the HVC model and evaluation of learned plasticity under synaptic turnover, as noted in the overall response.
>
> We have also clarified the experimental predictions that stem from our improved model of HVC. The work makes an explicit prediction for the nature of the learning rule in HVC that could be tested using methods to infer learning rules from data (Pereira and Brunel, 2018). Further, the hypothesis that plasticity on inhibitory and excitatory synapses might maintain different aspects of HVC(RA) cell firing leads to several predictions: (1) inhibition plays an essential role in maintaining the firing profile of HVC(RA) neurons, (2) manipulating the firing rate envelope of an HVC(RA) cell should cause the neuron to change both its excitatory and inhibitory inputs.
>
> We have simplified Fig. 1 in accordance with the reviewer’s comments: we removed the right portion of Fig. 1C and provided a more useful caption for the network diagram. We have additionally modified Fig. 2c to only include one trial. We have also standardized fonts and font sizes in all figures. We have further improved figure clarity by ensuring all symbols and colors are properly defined.
>
> We have rewritten the equation on L77 as suggested. It now reads:
>
> “Each neuron fired according to $x\_{j}(t) = \\left[ V\_j(t) - b \\right]^+,$ where $V\_j(t)$ evolves via $\\tau\_m \\dot{V}\_j(t) = -V\_j(t) + \\sum\_i w\_{ij} x\_i(t)$.”
>
> We thank the reviewer for catching the spacing issue on page 11.

---

> ### Comment · Reviewer_4Ykq · 2024-08-12
>
> Thank you for answering my questions and addressing the suggested manuscript improvements. I am happy to maintain my original high score.

---

### Official Review · Reviewer_rSVd · 2024-07-26

**Soundness:** 2
**Presentation:** 2
**Contribution:** 2
**Rating:** 6
**Confidence:** 3

**Summary:**

This paper proposes a method for discovering plasticity rules in spiking neural networks to achieve sequence generation using both excitatory and inhibitory dynamics. Rules are parametrized with basis function. The biological approach in the model involves also considerations homeostasis and robustness to perturbation. The HVC region of the zebra finch was used as inspiration. The study shows interesting finding, including a self-discovered generalisation of the Oja rule, robustness to perturbation and enhanced stability, including homeostatic mechanisms.

**Strengths:**

- The method uses interesting biological concepts, including both excitatory and inhibitory signals, and local Hebbian-like rules.
- The study analyses various properties of the evolved neural circuits, including robustness to perturbation and robustness to synaptic turnover.
- The model includes the use of noise, an important factor that affects both biological and artificial systems.
- The computational study supports the idea, not new in biology, that both excitatory and inhibitory synapses are important in maintaining equilibrium.
- The method allows for testing of particular hypothesis, e.g., the utility of dense feedforward connectivity.

**Weaknesses:**

- Unclear novelty: there have been many studies that evolved plasticity rules using some form of evolutionary computation and biological plausible rules. In fact, evolution of Hebbian-like rules has been a popular area of research for decades. While this study advances many previous studies, I feel that the novelty and contributions in this area are not well highlighted with respect to existing studies.
- It is unclear to me how the findings can be used more broadly. I agree that it is interesting the CMA-ES and meta-learning can optimize plasticity rules to have interesting biological properties, but how can this finding be used? This is the most important concern I have: what impact can this paper and its discoveries have in the field?
- Figures' clarity: there are a lot of assumption on the meaning of symbols, colours, etc, in the figure. E.g., Fig 1c, unless one is very familiar with the zebra fish, I don't think these graphs very useful. What is the big blue dot on top of the red ones? what do the colour in the cell-index vs. time mean? what does the graph mean?
- The manuscript describes synaptic turnover starting from the abstract, but a search of the term to find out what that is and how it is implemented did not provide satisfactory results.
- There may be an issue with the submission as the Supplementary information is not included, resulted also in broken references (e.g. line 197, page 7)

**Questions:**

- How many fitness evaluation through CMA-ES were performed? How much time did one evaluation take? How did the fitness evolve over time?
- Following up from a point in the weakness. How can these findings be used? Is there a recommendation that we should use evolution to implement and tune artificial spiking neural networks?

**Limitations:**

The paper addresses the limitations, in particular in relation to computational demands.

---

> ### Author Rebuttal · Authors · 2024-08-07
>
> We thank the reviewer for their careful consideration of our manuscript.
>
> We address the reviewer’s concerns regarding novelty in the common response. Briefly, our unique contribution is the exploration of unsupervised and unrewarded plasticity via meta-learning that organizes and maintains a specific and biologically relevant computational motif, a sequence. We have clarified this in the introduction. While prior work on sequence self-generation used hand-built rules, we learn the optimal plasticity rule by maximizing its ability to represent time within the network dynamics. Further, we believe the inclusion of biological noise, such as instability in synaptic connectivity, in a learning procedure is novel, particularly in the sequence generation context.
>
> Regarding broader impact, our work suggests a specific role for unsupervised plasticity: the establishment and maintenance of simple computational motifs. We hypothesize that the plasticity that organizes these computational motifs might be extremely specific to the structure being organized and may altogether lack feedback modulation. This suggests we should look for plasticity rules that might underlie the self-organization of other common computational motifs, such as line and ring attractors. From the standpoint of experimental predictions, this work provides a hypothesis for why behaviors like path integration may be innate to or rapidly learned by some species. It additionally suggests that such simple computation motifs in the brain may lack dopaminergic or other reward-related inputs.
>
> We apologize for the unclear definition of synaptic turnover: by this we mean the disappearance of existing synaptic connections and emergence of nascent ones over time. We now explicitly define it, and include an equation in Sec. 2.2. Our addition reads:
>
> “We next asked whether ongoing disruptions to network structure alter which plasticity rules are meta-learned. To explore this, we introduced synaptic turnover, a stochastic process by which existing neurons disappeared and new, small synapses emerged, to the simulation phase of the meta-learning loop. Prior to each network activation, all connections were updated according to
> $$    w_{ij} \\leftarrow \\begin{cases}
>       0 \\quad \rm{ if } \\quad  |w_{ij}| > 0 \\; \\mathrm{and} \\; x_{\rm ST} < p_{\rm ST}, \\\
>       \\; \\;w_{ij} \\quad \rm{ if } \\quad  |w_{ij}| > 0 \\; \\mathrm{and} \\; x_{\rm ST} \geq p_{\rm ST}, \\\
>       \\; \\;\epsilon \\quad \rm{ if } \\quad |w_{ij}| = 0 \\; \\mathrm{and} \\; x_{\rm ST} < p_{\rm ST},\\\
>       \\; \\; 0 \\quad \rm{ if } \\quad  |w_{ij}| = 0 \\; \\mathrm{and} \\; x_{\rm ST} \geq p_{\rm ST}\\}
>     \\end{cases}
> $$
> where $x_{ST} \sim U[0, 1]$, $p_{ST}$ dictates the rate of synaptic turnover, and $\epsilon$ is a small positive (negative) constant if the presynaptic cell is excitatory (inhibitory). Since per Eq. 1, synaptic turnover determines the set of synapses available to the plasticity rule since the rule is unable to act on connections of size 0. To ensure learned rules were robust to a spectrum of rates of synaptic turnover, only half the networks used to evaluate the batch loss underwent this process (Fig.1f, bottom)."
>
> Note, OpenReview does not seem to render \begin{cases} within equations correctly.
>
> We have added the following description of the computational resources used for this study to the discussion:
>
> “Training E$\rightarrow$E plasticity across 10 networks typically required 24 hours of compute on 30 Cascade Lake or Ice Lake Intel CPU cores to yield reasonable solutions. Training plasticity across all sets of synapses as in Sec. 2.3 required up to 72 hours across the same number of cores.”
>
> Following the reviewer’s suggestion, we have added a section with background on zebra finch physiology. This section reads:
>
> “In the zebra finch, premotor nucleus HVC contains excitatory neurons that fire sparsely (typically in one burst of spikes) during song and are purportedly arranged in a feed-forward structure (Fig. 1c). A subset of these cells, known as $\\mathrm{HVC}\_{\rm (RA)}$ neurons, project to downstream nucleus RA (robust nucleus of the archistriatum), which in turn projects to vocal neurons of the syrinx and to the brainstem, which regulates respiration \cite{Hahnloser_2002}. HVC additionally receives excitatory projections from nucleus Uva, which controls the onset of song syllables \cite{Moll2023} and provides input for the duration of song \cite{Danish2017}. $\mathrm{HVC}_{\rm (RA)}$ neurons inhibit each other disynaptically via a population of inhibitory interneurons within HVC \cite{Kosche_2015}.”

---

> > ### Comment · Reviewer_rSVd · 2024-08-08
> >
> > The authors made significant efforts to address my comments, and have also improved the paper according to the other reviewers' comment. I'm therefore happy to increase my evaluation.

---

### Official Review · Reviewer_JAan · 2024-07-31

**Soundness:** 3
**Presentation:** 3
**Contribution:** 3
**Rating:** 6
**Confidence:** 3

**Summary:**

The paper proposes a conceptually simple set-up to learn plasticity rules that result in neural networks that perform a specific task. The set-up of the paper is elegant and made up of simple but effective elements, such as linear-threshold neurons and evolutionary optimisation. The authors obtain some intriguing results, e.g. that their method discovers plasticity rules well-known in experimental neuroscience.

**Strengths:**

The paper tackles an interesting problem with a relatively straightforward set-up, which makes it easily understandable. It has a very high density of results, which are technically sophisticated as well as relevant for many questions in neuroscience.

**Weaknesses:**

One of the main limitations of the paper is that this very promising and flexible set-up is only applied to one very idiosyncratic task of elapsed time decoding. This is a very specific task, and not one that brains do spontaneously (see next paragraph), so although it yields interesting results it seems a bit lacking. It would be tremendously useful to see the results in a broad range of tasks, to see e.g. under what conditions other common plasticity rules (Hebbian, anti-Hebbian, perhaps some dopamine-like reward-modulated plasticity) emerge.

The motivation of the system set-up is somewhat dubious. The authors point out the importance of "intrinsic, self-organised dynamics" but then include as the primary component of their loss function the performance of an external decoder that predicts simulation time. Neither the decoder nor the ground truth time values are in any way intrinsic to neural activity, so I would argue this is not representative of self-organisation.

On a separate note, the paper sorely lacks any demonstration of the "downstream" effects of these plasticity rules. For example, the author's motivation is that "self-organized computations provide scaffolds for solving difficult tasks". However, the authors don't actually show that the networks evolved with their discovered plasticity rules can actually solve any difficult task.

The paper gets really hard to follow towards the end, particularly Sections 2.3.2 and 2.3.3. The intervention to increase or decrease E afferents in Sec. 2.3.2, as well as the analysis metrics in Fig. 6a-c aren't very clear. It also surprising to see Sec. 2.3.3, the final results subsection of the paper, introducing a whole new model and a different plasticity rule, which are only partially related to the model and plasticity rules studied elsewhere. Overall, I could not understand well the set-up or results in either of these subsections.

The details of the meta-learning procedure should be explained more thoroughly in Sec. 2.1.1. Figures are quite busy and in general hard to parse.

**Questions:**

The p-values in Sec. 2.2.1 are somewhat meaningless, because one can always run more simulations to decrease the p-value. I would recommend quantifying effect size (with e.g. Cohen's $d$) instead.

Please define important terms at first use (e.g. HVC).

What is $\mathcal{P}(\textbf{C})$ in L. 103?

References 4, 5, 11 and 28 are misformatted.

**Limitations:**

The "Discussion and limitations" section is very short and should be expanded.

---

> ### Author Rebuttal · Authors · 2024-08-07
>
> We thank the reviewer for their careful reading of our paper and their comments.
>
> We agree with the reviewer that this promising methodology should be used to study the self-organization of other computational motifs, such as line and ring attractors, and we hope that this paper provides a path for doing so. We have added a line in the introduction highlighting this potential future direction. It reads:
>
> “Though the results we present in this work are specific to sequence generation, our methodology might be applied to other basic circuit motifs such as line and ring attractors.”
>
> We also agree the approach should be expanded in the future to include different types of plasticity, including feedback-based rules, while this is not entirely new: Shervani-Tabar and Rosenbaum (Nat. Commun., 2023) learn feedback-based, biologically plausible rules for deep network training. We view our work as a proof of principle for studying the self-organization of robust computational motifs.
>
> We agree that neurons in general have no knowledge of ground truth simulation time. Time in our simulation is locked to the time of an input. We have clarified in our methods that we divide the simulation of each plasticity rule into 400 activations of 110 ms. At t = 10 ms of each activation, the initial kick of excitation is presented to the network. Since this stimulus is time-locked to the simulation clock, attempting to decode time elapsed since the beginning of the activation is the same as decoding the time elapsed since stimulus presentation. Thus, our networks track the time since the inciting pulse initiates the dynamics, and the decoder is merely a means by which to evaluate the success of the plasticity rule at accomplishing this.
>
> We have rewritten section 2.1.1 to better describe the meta-learning procedure, as requested. It now reads:
>
> “We next define a loss function that evaluates the quality of the sequential dynamics organized under a chosen $\\{c\_k, \tau\_k\\}$, and attempt to minimize it using an evolutionary strategy, Covariance Matrix Adaptation (CMA-ES)\cite{auger_and_hansen}. CMA-ES places a multivariate Gaussian on the space of the parameters to be optimized, then samples from that Gaussian and evaluates the fitness at each point. The fitness is then used to adjust the mean and covariance matrix of the distribution in an attempt to iteratively maximize fitness. We use CMA-ES to sample from the space of possible $\\{c\_k, \tau\_k\\}$. We evaluate the loss for each choice of values $\\{c\_k, \\tau\_k\\}$ defining the plasticity rule by simulating 10 randomly initialized networks under this rule and evaluating the resultant dynamics at the end of the simulation. Each simulation consists of 400 activations of 110 ms. At $t = 10$ ms of each activation, an arbitrary neuron is driven by a strong kick of excitation \cite{tupikov_2021_addition}. Following this, all other neurons in the network receive Poisson distributed input for a period of $100$ ms. We generate the Poisson inputs as a sum of a component which is generated randomly trial by trial, and a frozen component, held fixed from trial to trial, which we take to model song-locked input to HVC from the nucleus Uva \cite{Murray_2017}. The total loss for a given rule is simply the sum of losses across each of the 10 networks.”
>
> We have attempted to declutter figures and enlarge the size of figure labels and text, where possible.
>
> We have also clarified sections 2.3.2 and 2.3.3 to more clearly describe our procedure and results in perturbing networks organized under joint excitatory and inhibitory plasticity.
>
> As requested, we have added a quantification of Cohen’s $d$ to Sec. 2.2.1. We found that the learned rules have a moderate to large effect on the loss when compared against the rule from the literature. We report this in the text as follows:
>
> “We used the Kruskal-Wallis $H$ test to test for equality of medians. We found meta-learned rules trained with and without synaptic turnover outperformed the rule based based on rigid synapse constraints ($p=2.5 \, \mathrm{x} \, 10^{-8}$, Cohen's $d = -0.45$ and $p=7.2 \, \mathrm{x} \, 10^{-7}$, Cohen's $d = -0.44$, respectively), while in the absence of perturbation, medians were not distinct after 4-fold Bonferroni correction ($p=0.016$, Cohen's $d = -0.29$  and $p = 0.029$, Cohen's $d = -0.30$, respectively).”
>
> We have added a definition of HVC, which historically was called the High Vocal Center, but is now simply used as a proper noun. We include a description of HVC physiology in the response to R2.
>
> We have also amended $\mathcal{P}(\textbf{c})$ on L. 103 to $\mathcal{P}(\textbf{c}, \tau)$ (note: $\tau$ should be bolded as well). $\mathcal{P}(\textbf{c})$ has no definition; this was a typo.
>
> We thank the reviewer for pointing out our misformatted references.

---

> > ### Comment · Reviewer_JAan · 2024-08-12
> >
> > Many thanks to the authors for their rebuttal. I remain unconvinced that the method is in any way intrinsic (time-since-input is functionally the same as time-since-simulation-start), but I still think it's an interesting set of experiments.

---

### Author Rebuttal · Authors · 2024-08-07

We thank all reviewers for their thoughtful feedback. We appreciate their overall support of the manuscript and their constructive criticism. Below, we address their major concerns.

R1 and R4 noted that the paper lacked a demonstration that the learned plasticity rule accelerated the ability of the self-organized sequential dynamics to support supervised learning of a downstream output pattern. We chose to focus on the self-organization of the timing scaffold as the utility of a timing representation to support motor learning has already been well studied (e.g. Duffy et al. PNAS 2019). Nicola and Clopath (Nat. Commun. 2017) demonstrated that providing a stable representation of time (i.e. a sequential input) accelerates a network’s ability to perform a birdsong-like motor task (see section “High dimensional temporal signals improve FORCE training”). Fiete et al. (J. Neurophys. 2004) argued that temporal sparseness in the premotor drive of HVC vastly simplifies the task of producing some sequential motor output by decomposing it into the approximately independent subtasks of producing the correct motor output at each moment in time; this was also demonstrated in a biologically plausible implementation of RL learning of the same task (Farries and Fairhall J. Neurophys. 2007). Lastly, providing a time input to agents has become popular in the reinforcement learning literature as it permits an agent to adjust its policy in accordance with elapsed time in the task (Wang et al. arXiv preprint arXiv:1611.05763 2016), suggesting another downstream benefit of time representation that deserves further exploration.

Regarding concerns of unclear novelty raised by R2, R3, and R5, we acknowledge this paper builds upon a rich body of work that attempts to evolve Hebbian-like rules via a supervised procedure. We now clarify in the introduction that our unique contribution is the exploration of unsupervised and unrewarded plasticity via meta-learning that organizes and maintains a specific and biologically relevant computational motif, a sequence. While prior work on sequence self-generation used hand-built rules, we learn the optimal plasticity rule by maximizing its ability to represent time within the network dynamics. Further, we believe the inclusion of biological noise, such as instability in synaptic connectivity, in a learning procedure is novel, particularly in the sequence generation context. This question is motivated by recent neuroscience findings about representational drift. Our work illustrates how the inclusion of such noise can shift the optimal learning rule, and demonstrates how plasticity responsible for self-organization can coexist with maintenance mechanisms in a single rule, which to our knowledge is also novel in the context of sequences.

We also point out that this work is novel in its application of meta-learning of plasticity rules to a problem in which the model network is highly recurrent, the rule basis is large, and the optimal rule is not known a priori. Prior work has largely focused on feedforward structures whose outputs can be computed quickly, streamlining the meta-learning process (Confavreux et al. NeurIPS 2020, Shervani-Tabar and Rosenbaum Nat. Commun. 2023, Lindsey and Litwin-Kumar NeurIPS 2020), rule bases that were relatively small (Shervani-Tabar and Rosenbaum Nat. Commun. 2023, Tyulmankov Neuron 2022), or problems where the optimal plasticity rule was already known (Confavreux et al. NeurIPS 2020). Here we show that meta-learning can yield successful rules that are dense in the given basis, requiring us to develop additional techniques to understand their function. Such an example is important to include in the literature for future authors intending to use meta-learning. We also believe the inclusion of an L1-like penalty on synaptic change is novel and represents a more biologically realistic constraint on a plasticity rule than penalizing term coefficients, as other studies have done.
R1 and R4 also noted section 2.3, “Including inhibitory plasticity,” was difficult to parse. We have rewritten this section to clarify our findings from the rule perturbation analysis and provide more detail regarding the single neuron manipulations we performed (see individual responses).

At the request of R2 and R5, we have added background on zebra finch physiology (see response to R5) that describes the constituent cell types and connectivity within sequence-generating nucleus HVC and the role the nucleus is thought to play in song production.
Regarding broader impact, our work suggests a specific role for plasticity that is unmodulated by feedback: the setup of simple computational motifs. We hypothesize that the plasticity that organizes these computational motifs might be extremely specific to the dynamics being organized and may not require supervision. We feel that this work will inspire future work to seek plasticity rules that underlie the self-organization of other common computational motifs, such as line and ring attractors. From the standpoint of experimental predictions, this work provides a hypothesis for why behaviors like path integration may be innate to some systems. It additionally suggests that the establishment of such simple computational motifs in the brain may not require dopaminergic or other reward-related inputs.

---

### Decision · Program_Chairs · 2024-09-25

**Decision:**

Accept (poster)

**Comment:**

This paper proposes a procedure for discovering the synaptic plasticity rules by which a neural network can learn sequences. There were some questions as to the novelty and generalizability of this work; however, there was broad agreement that this work represents an interesting contribution to the field of theoretical neuroscience. Therefore, I recommend acceptance.